# A Coupled Heat Transfer Calculation Strategy for Composite Cooling Liquid Rocket Engine

**Bo Xu** [1], **Bing Chen** [1,*], **Jian Peng** [1], **Wenyuan Zhou** [1,2] **and Xu Xu** [1]

[1] Department of Astronautics, Beihang University, Beijing 100191, China; xubo2022@buaa.edu.cn (B.X.)
[2] Shanghai Institute of Space Propulsion, Shanghai 201112, China
\* Correspondence: markchien@buaa.edu.cn

**Abstract:** To better understand the characteristics of coupled heat transfer in liquid rocket engines, a calculation scheme is proposed in this paper. This scheme can simulate the coupled heat transfer processes, including combustion and flow in the thrust chamber, radiation heat transfer, heat conduction in the wall, heat transfer of coolant flow in the cooling channel, and gas film cooling in the thrust chamber wall. The numerical method used in each physical area, the data transfer method between each computing module, the strategy of data transfer on the coupling interface, the calculation process, and the convergence criterion are all introduced in detail. The calculation scheme was verified by analyzing a water-cooled nozzle. Then, the coupled heat transfer calculation was carried out for a liquid rocket engine using a propellant composed of unsymmetrical dimethylhydrazine and dinitrogen tetroxide. Two working conditions were analyzed: whether the gas film cooling was performed or not. The results showed that the algorithm successfully indicated the protective effect of the gas film on the wall surface, and the calculation results were reasonable. It played a guiding role for the coupled heat transfer of the liquid rocket engine using a composite cooling method.

**Keywords:** liquid rocket engine; coupled heat transfer; regenerative cooling; film cooling

## 1. Introduction

Faced with growing demands for better performance, the liquid rocket engine is designed to withstand extreme pressure and temperature, which poses a greater challenge to the reliability of thrust chambers. At present, the combination of active cooling and passive cooling is recognized as an effective cooling scheme in engineering [1]. Regenerative cooling is the standard cooling system for almost all modern main-stage, booster, and upper-stage engines [2]. Other cooling methods, such as film cooling, transpiration cooling, ablative cooling, radiation cooling, heat sink cooling, and dump cooling, are also used to reduce the regenerative cooling load and propellant demand [3]. The high manufacturing cost of liquid rocket engines and the toxicity of hypergolic propellants limit the ground test to some extent. Therefore, using numerical simulation to predict the performance of the engine can greatly shorten the development cycle and reduce the development cost.

Heat transfer in the thrust chamber involves many physical processes, including combustion and flow in the thrust chamber, the convection and radiation heat transfer of high-temperature gas to the wall, heat conduction inside the wall, heat transfer between the coolant and wall in the cooling channels, radiation heat transfer between the outer wall and the surrounding environment, and so on, which is a typical fluid–structure heat transfer problem.

For the coupled heat transfer problem, the first method is called the fully coupled method, where the same extended CFD code is used for both the fluid and solid regions. Earlier, Han et al. [4] used this method to study the heat transfer of turbomachinery. However, the time scale difference between fluid convection and solid heat conduction is huge. Therefore, in some cases, the stability of this method has problems, and the solution of the fluid and solid needs to be decoupled [5].

The second method, the loosely coupled method, uses a special heat conduction program to solve structural heat transfer and a CFD program to solve flow. To simplify the implementation, the loosely coupled method between fluid and solid solvers is often adopted [6]. When the loosely coupled method is adopted, the convergence of calculation depends on the method of information exchange between the fluid regions and the solid regions. Currently, there are four commonly used methods. The first is the Flux Forward Temperature Backward (FFTB) method [7], in which the heat flux boundary condition is adopted in the solid regions, while the temperature boundary condition is adopted in the fluid regions. The Temperature Forward Flux Backward (TFFB) method [8] does the opposite. The other two methods are the heat transfer coefficient Forward Temperature Backward (hFTB) method [9] and the heat transfer coefficient Forward Flux Backward (hFFB) method [10]. These two methods use convective heat transfer coefficient and reference temperature to play the role of heat flux. These methods were reviewed in the literature [10]. Recently, Remiddi et al. [11] used the hFTB method to calculate the coupled heat transfer in the rocket engine.

When the loosely coupled method is adopted, the structural wall mesh and the fluid mesh are often not matched, and data transfer is required at the coupling interface. Multiple interpolation methods can accomplish this task. Han et al. [12] used the linear interpolation method to exchange data in the combustion chamber and wall coupling region when calculating the coupled heat transfer of the attitude control liquid rocket engine. Wang et al. [13] proposed a parallel multistep radial basis function interpolation method. This method is efficient and stable in dealing with complex geometric deformation.

The accurate prediction of the flow and heat transfer process in the thrust chamber is the premise of the coupled heat transfer calculation of the liquid rocket engine. Zhang et al. [14] developed a liquid rocket engine two-phase combustion and flow solver. It considered the breakup and secondary breakup of the liquid jet in the turbulent flow field, as well as the physical processes such as the interaction between the droplet and the gas-phase turbulence, droplet evaporation, and gas-phase combustion. The relationship between gas temperature, pressure, reaction product type, specific impulse, and combustion efficiency in the combustion chamber was discussed. In terms of film cooling, Shine's research [15–17] showed that the injector structure has an important impact on film cooling, and there is an optimal blowing ratio for a given geometry. Xiang et al. [18–20] studied the effect of gas film cooling holes with different angles on the cooling effect in rocket engines with various propellant combinations through experiments and numerical simulation. Fu et al. [20] developed a droplet/wall impact model which performed better than the O'Rourke and Amsden model. In terms of regenerative cooling, Ulas et al. [21] studied the temperature and pressure drop characteristics of the cooling channel of the LOX/kerosene liquid rocket engine by changing the aspect ratio of the cooling channel and the cooling channel area of the non-critical region. Pizzarelli et al. [22,23] analyzed the influence of the aspect ratio of the cooling channel on the coupled heat transfer and established a quasi-two-dimensional model of supercritical coolant which suits a cooling channel with a large aspect ratio. In terms of radiation, Abdelaziz [24] studied the radiation distribution in the combustion chamber of a Rolls Royce RB-183 turbofan engine and the difference between the Discrete Ordinates Method (DOM) and the P-1 method in radiation calculation. The research showed that the DOM method is more accurate in the estimation of combustion chamber radiation. Sun et al. [25] proposed a hybrid method combining the lattice Boltzmann method with the finite volume method. This method can solve the combination problem of radiation and heat transfer under irregular geometry circumstances. In addition, the combustion mechanism of hypergolic propellant is relatively complex. Taking the reaction of MMH and NTO as an example, the reaction includes the sub-mechanism of MMH decomposition, NTO thermal decomposition, MMH/NTO and intermediates, and small hydrocarbons [26]. Hou et al. [27,28] proposed the chemical reaction mechanism of the MMH/NTO reaction, including 20 reactions and 23 components.

At present, many researchers have conducted relevant studies on the coupled heat transfer of liquid rocket engines [29–34]. In the literature we have reviewed, there is relatively little research on the coupled heat transfer mechanism of multiple physical regions in liquid rocket engines. The mechanism of data transfer between different computing regions, the solution flow chart of the coupled heat transfer problem, convergence criteria, and so on, are lacking.

In this paper, a platform with strong extensibility and high stability for the heat transfer calculation of composite cooling liquid rocket engines is provided, and a corresponding numerical method is presented. The three-dimensional Navier–Stokes (N-S) equation is discretized by the density-based finite volume method to solve gas flow. The chemical equilibrium laminar flamelet model was used to calculate the turbulent combustion in the thrust chamber. The SIMPLE algorithm was used to solve the turbulent flow of the coolant. The heat conduction differential equation was discretized by the finite element method to calculate the wall temperature. The radiation transfer equation was solved by the finite volume method to calculate the gas radiation in the thrust chamber. All these numerical codes were loosely coupled with an emphasis on the data transfer strategy, calculation process, and convergence criterion. The coupling strategy effectiveness was verified by the Arnold Engineering Development Center high-enthalpy nozzle flow. Finally, the coupled heat transfer calculation of an unsymmetrical dimethylhydrazine (UDMH)/dinitrogen tetroxide (NTO) liquid rocket engine with regenerative cooling, film cooling, and radiation cooling was carried out. The numerical results of wall temperature and heat flux of the thrust chamber with and without gas film are discussed in depth in the following sections.

## 2. Governing Equation and Numerical Algorithms

### 2.1. Flow and Combustion

In this paper, the three-dimensional N-S equation is used to solve the gas flow in the thrust chamber, and the laminar flamelet model is used to model turbulent combustion. The governing equations of the flow field include the mass conservation equation, momentum conservation equation, energy conservation equation, and the equation of mixture fraction and its variance. The system is expressed as:

$$\frac{\partial \boldsymbol{U}}{\partial t} + \nabla \cdot [\boldsymbol{F}(\boldsymbol{U}) - \boldsymbol{G}(\boldsymbol{U})] = \boldsymbol{S} \tag{1}$$

where $t$ is the time, and $\boldsymbol{U}$, $\boldsymbol{F}(\boldsymbol{U})$, $\boldsymbol{G}(\boldsymbol{U})$, and $\boldsymbol{S}$ are the conservation variables, inviscid flux, viscous flux, and source term, respectively. Their vector forms are given by:

$$\boldsymbol{U} = \left[\rho, \rho u_1, \rho u_2, \rho u_3, \rho e, \rho z, \rho z^{''2}\right]^T \tag{2}$$

$$\boldsymbol{F}(\boldsymbol{U}) = \boldsymbol{F}_1 e_1 + \boldsymbol{F}_2 e_2 + \boldsymbol{F}_3 e_3 = \boldsymbol{F}_i e_i \tag{3}$$

$$\boldsymbol{G}(\boldsymbol{U}) = \boldsymbol{G}_1 e_1 + \boldsymbol{G}_2 e_2 + \boldsymbol{G}_3 e_3 = \boldsymbol{G}_i e_i \tag{4}$$

where $\rho$ is the density, $e_i$ ($i$ = 1,2,3) is the unit vector in Cartesian coordinates $x_i$ ($i$ = 1,2,3), $u_i$ is the velocity in the $x_i$ direction, $e$ is the specific total energy, $z$ is the mixture fraction, and $z^{''2}$ is the variance in the mixture fraction. The expression of inviscid flux and viscous flux is:

$$\boldsymbol{F}_i = \begin{pmatrix} \rho u_i \\ \rho u_i u_1 + p\delta_{i1} \\ \rho u_i u_2 + p\delta_{i2} \\ \rho u_i u_3 + p\delta_{i3} \\ \rho u h \\ \rho u z \\ \rho u z^{''2} \end{pmatrix}, \boldsymbol{G}_i = \begin{pmatrix} 0 \\ \tau_{i1} \\ \tau_{i2} \\ \tau_{i3} \\ \tau_{ij} u_j - q_i + \varpi_i(k) \\ \varpi_i(z) \\ \varpi_i(z^{''2}) \end{pmatrix} \tag{5}$$

where $\delta_{ij}$ is the Kronecker operator, and $p$ and $h$ are the pressure and total enthalpy per unit mass, respectively. Shear stress tensor $\tau_{ij}$, heat flux $q_i$, and operator $\varpi$ are expressed as:

$$\tau_{ij} = (\mu + \mu_t)\left(\frac{\partial u_i}{\partial x_j} + \frac{\partial u_j}{\partial x_i} - \frac{2}{3}\delta_{ij}\frac{\partial u_m}{\partial x_m}\right) \tag{6}$$

$$q_i = -\left(\frac{\kappa}{c_p} + \frac{\mu_t}{\mathrm{Pr}_t}\right)\frac{\partial h}{\partial x_i} + \sum_{n=1}^{N_s}\left(\rho D_n + \frac{\mu_t}{Sc_t}\right)h_n\frac{\partial Y_n}{\partial x_i} \tag{7}$$

$$\varpi_i(k) = \left(\mu + \frac{\mu_t}{\sigma_k}\right)\frac{\partial k}{\partial x_i} \tag{8}$$

$$\varpi(z) = \left(\rho D + \frac{\mu}{Sc_t}\right)\frac{\partial z}{\partial x_i} \tag{9}$$

$$\varpi(z''^2) = \left(\rho D + \frac{\mu_t}{Sc_t}\right)\frac{\partial z''^2}{\partial x_i} \tag{10}$$

where $\mu$ is the viscosity coefficient, Pr is the Prandtl number, Sc is the Schmidt number, and the subscript $t$ denotes turbulence. Turbulent parameters are determined by the SST $k - \omega$ model. Among the properties used in this paper, the thermal conductivity $\kappa_i$ and dynamic viscosity coefficient $\mu_i$ are determined by Eucken's formula [34] and the Lennard-Jones formula [35], respectively. The properties of the mixtures, such as viscosity coefficient $\mu$, diffusion coefficient $D$, and heat conduction coefficient $\kappa$, are calculated by Wilke's mixing rules. Eucken's formula and the Lennard-Jones formula are presented in Appendix A.

The first six equations in Equation (1) have a source term of zero, and $s_7$ is expressed as:

$$s_7 = 2\frac{\mu_t}{Sc_{t2}}(\nabla z \cdot \nabla z) - \rho\chi \tag{11}$$

The $\chi$ is the scalar dissipation rate, which measures the deviation from the equilibrium state. As the laminar flamelet model based on chemical equilibrium is adopted in this paper, this term is zero. The values of the constants in the equations are shown in Table 1.

**Table 1.** Constants in the equations.

| $\mathrm{Pr}_t$ | $Sc_t$ | $Sc_{t2}$ | $\sigma_k$ |
|---|---|---|---|
| 0.9 | 0.5 | 0.5 | 1.0 |

In this paper, the finite volume method is used to discrete the N-S equation. The HLLC scheme [36] is used for the spatial discretization of the circulating flux. The standard center scheme was used for the discretization of viscous flux. The implicit lower–upper symmetric Gauss–Seidel (LU-SGS) scheme [37] was used to solve the above equations. The UDMH/NTO two-parameter indexed instantaneous laminar flamelet table $Y_i(z,z''^2)$ was generated by the Fluent software. The concentration of each component in the flow field was obtained by the flamelet table. Temperature was obtained by solving the energy equation, not the flamelet table. In other words, the flamelet table is used as a complex equation of state to describe the relationship between local temperature and composition, density, pressure, etc. [38]. The boundary conditions at the propellant inlet are expressed as:

$$\begin{cases} z = 0, z''^2 = 0, Y_i = Y_{i,O} \\ z = 1, z''^2 = 0, Y_i = Y_{i,F} \end{cases} \tag{12}$$

### 2.2. Solid Heat Conduction

The thermal conduction differential equation was used to describe the heat conduction inside the wall, which leads to the following expression:

$$\rho c \frac{\partial T}{\partial t} = \frac{\partial}{\partial x}(\lambda_x \frac{\partial T}{\partial x}) + \frac{\partial}{\partial y}(\lambda_y \frac{\partial T}{\partial y}) + \frac{\partial}{\partial z}(\lambda_z \frac{\partial T}{\partial z}) + Q_v \tag{13}$$

where $\rho$ is the density of the structure material, $c$ is the specific heat capacity, $T$ is the temperature, $t$ is the time, $\lambda_i (i = 1, 2, 3)$ is the conductivity in the $x_i$ direction, and $Q_v$ is the influence of the heat source in the structure.

Through the Galerkin weighted parameter method, the equation was discretized and solved by the finite element method.

### 2.3. Regenerative Cooling

The SIMPLE algorithm was used to solve the approximately incompressible flow of coolant in the cooling channel. The general governing equation of this algorithm is:

$$\frac{\partial(\rho\varphi)}{\partial t} + \nabla \cdot (\rho u \varphi) = \nabla \cdot (\Gamma_\varphi \nabla \varphi) + S_\varphi \tag{14}$$

where $\varphi$ is an arbitrary variable, and it takes 1, $u_i$, and $h$ to correspond to the continuity equation, the momentum equation in the $x_i$ direction, and the energy equation, respectively. $\Gamma_\varphi$ is the generalized diffusion coefficient. For the continuity equation, the momentum equation, and the energy equation, the values are 0, $\mu_{eff} = \mu + \mu_t$, and $\lambda_{eff} = c_p(\mu/\text{Pr} + \mu_t/\text{Pr}_t)$, respectively. $S_\varphi$ is the generalized source term.

### 2.4. Radiation of Gas

In the calculation of the gas radiation in the thrust chamber of the liquid rocket engine, only three atomic gases with strong radiation capacity, such as $CO_2$ and $H_2O$, were considered. Since the wall of the engine thrust chamber is heated at a high temperature and emits radiant energy, to obtain the radiation heat flux of the wall of the thrust chamber, both gas and wall should be used as the radiation source for the calculation. The finite volume method [39] was used to calculate the radiation heat transfer of both gas and wall in the thrust chamber of the liquid rocket engine. Ignoring gas scattering, we can finally establish the equation:

$$\int_{\Omega^m} \int_{A_i} I_{k\lambda,c}^m (s^m \cdot n_i) dA_i d\Omega^m = \int_{\Omega^m} \int_{V_p} [-\beta_k I_{k\lambda}^m(s) + \kappa_k I_{bk\lambda}(s)] dV_p d\Omega^m \tag{15}$$

where $\Omega^m$ is the solid angle, $I_{k\lambda,c}^m$ is the spectral radiation intensity in the solid angle $\Omega^m$, $s^m$ is the unit normal vector in the solid-angle direction, $A_i$ is the control body surface area, $n_i$ is the unit normal vector of the control body boundary surface, $V_p$ is the control body volume, and $\kappa_k$ is the absorption coefficient of the spectrum band.

### 2.5. Outer Wall Radiation

The extension section of the nozzle and the thrust chamber of the low-thrust engine with high-temperature-resistant materials usually adopt the radiation cooling method with a simple structure. That is, the heat flow is transferred from the combustion products to the wall of the thrust chamber, and then from the outside wall of the thrust to the surrounding environment in the form of radiation heat dissipation. For radiative cooling, the heat flux is calculated using the Stefan–Boltzmann law:

$$q = \varepsilon \sigma T^4 \tag{16}$$

where $\varepsilon$ is the surface emissivity of the outer wall, $\sigma$ is Boltzmann's constant, and $\sigma = 5.67 \times 10^{-8} \text{ W}/(\text{m}^2 \cdot \text{K}^4)$.

### 3. Coupled Computing Strategy

*3.1. Data Transfer Policy*

Figure 1 shows the data transfer between different computing regions. The principle of data exchange is to ensure that the temperature and heat flux of the fluid and solid at the coupling interface remain continuous. It can be expressed as:

$$T_{solid} = T_{fluid}, q_{solid} = q_{fluid} \tag{17}$$

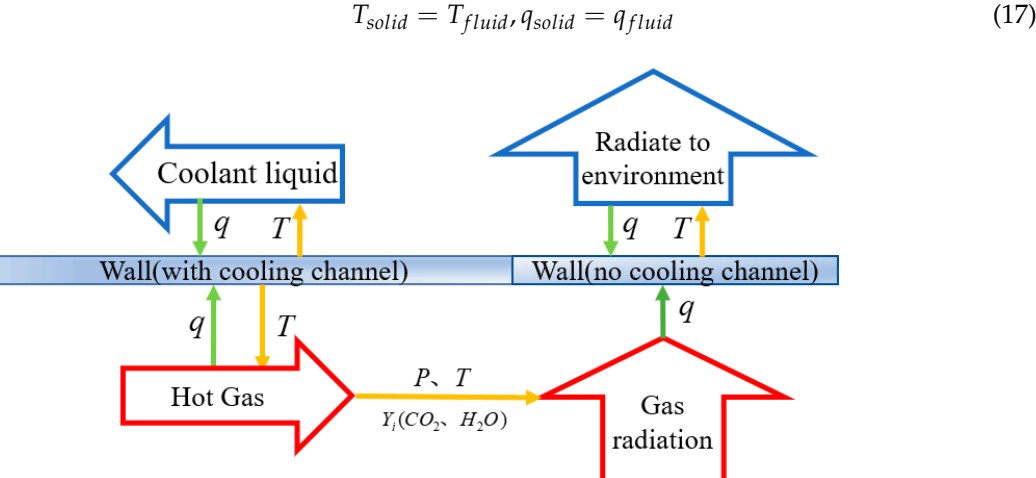

**Figure 1.** Data transfer between codes.

For solid and fluid regions, the boundary conditions used at the coupling interface are different. The temperature boundary condition is used in the fluid regions such as the flow field in the thrust chamber and the regenerative cooling channel, and the heat flux boundary condition is used in the solid regions. The convective heat flux on both sides of the wall is calculated by the fluid solvers. Then, the gas radiation module reads the temperature, pressure, and mass fraction of the $CO_2$ and $H_2O$ in the flow field and outputs the radiant heat flux. Then, the heat flux on both sides of the wall is input into the heat conduction module to calculate the temperature on both sides of the wall. The above steps are repeated until the calculation converges. To improve the stability of the calculation, the temperature and heat flux data transferred at the coupling interface are under-relaxed.

*3.2. Data Transmission of Unmatched Grids*

As shown in Figure 2, grids between structure and fluid are often mismatched. To achieve the numerical transfer of the two mesh interfaces, the three-dimensional linear interpolation method was adopted here. This method is simple to operate, has a clear physical meaning, and can achieve high precision when the grid is dense.

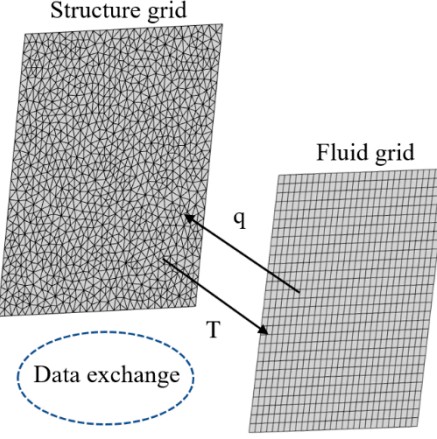

**Figure 2.** Data exchange between unmatched grids.

### 3.3. Flow Chart of Calculation

The calculation process is shown in Figure 3, where the orange line represents the heat flux value transferred between the coupling interfaces. The heat flux on the hot side includes two parts: convective heat flux and radiant heat flux. The former is obtain by the combustion and flow module. The latter is obtained by the radiation calculation module. The unsteady heat conduction module reads the heat flux on both sides of the wall, advances by a small time step, and outputs the temperature on both sides of the wall as boundary conditions for the calculation of the combustion and flow module and the regenerative cooling module, as shown by the green line. A basic assumption of coupled heat transfer is adopted here: the characteristic time of the flow field is much shorter than that of the heat conduction, so the heat conduction should be calculated under the condition that the flows on both sides of the wall converge. However, if the time precision is not pursued, the requirements for the step size of fluid can be relaxed.

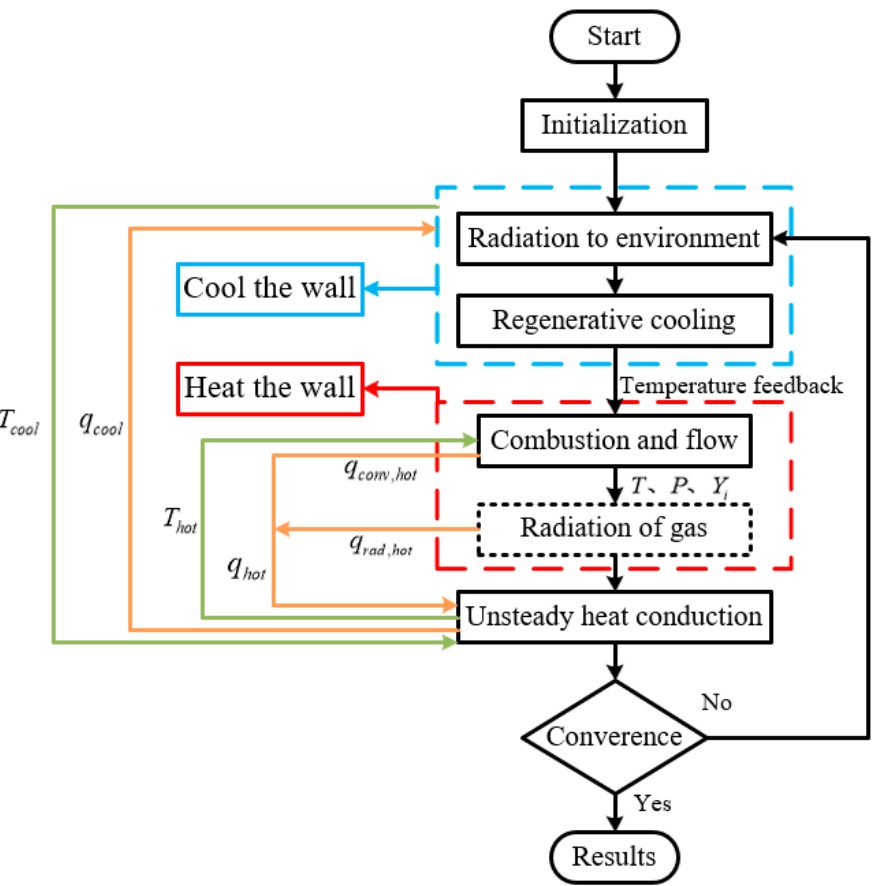

**Figure 3.** Flow chart of the loosely coupled calculation.

The temperature feedback step in the process refers to the calculation of the regenerative cooling module feeding back the coolant temperature at the liquid film injection point and the temperature when the coolant is injected into the combustion chamber to the combustion and flow module. This step can realize the dynamic adjustment of the propellant injection point temperature of the thrust chamber.

Because the calculation of gas radiation is time consuming, and the radiant heat flux is small compared to the convective heat flux, even if the radiant heat flux changes dramatically. Coupled with the convective heat flux, such drastic changes may be annihilated. Therefore, it is not necessary to calculate the radiation of the gas on the wall in every iteration, but rather every few iterations. This processing method can greatly reduce the calculation overhead and shorten the calculation time.

*3.4. Condition of Convergence*

When the wall is heated and cooled at the same time, the heat absorbed by the hot side is equal to the heat lost by the cold side when the heat transfer reaches a steady state. From the perspective of energy conservation, when the total heat flow on the hot side is equal to the total heat flow on the cold side, and the total heat flow remains unchanged during several iterations, the iterative calculation can be considered to have converged. The convergence criteria are as follows:

$$Q_{hot} = Q_{cool} \tag{18}$$

## 4. Results and Discussion

The feasibility and efficiency of the coupled heat transfer strategy are checked by two representative cases. The first case is the Arnold Engineering Development Center (AEDC) high-enthalpy nozzle flow, and the second case is a liquid rocket engine.

*4.1. AEDC High Enthalpy Nozzle*

To verify the effectiveness of the loosely coupled heat transfer algorithm, the AEDC high-enthalpy nozzle [40] was numerically simulated. The nozzle configuration is shown in Figure 4. The cooling channel adopts a ring slot structure. The cooling channel is divided into two sections, and the flow direction of the cooling water is from upstream of the nozzle to downstream of the nozzle.

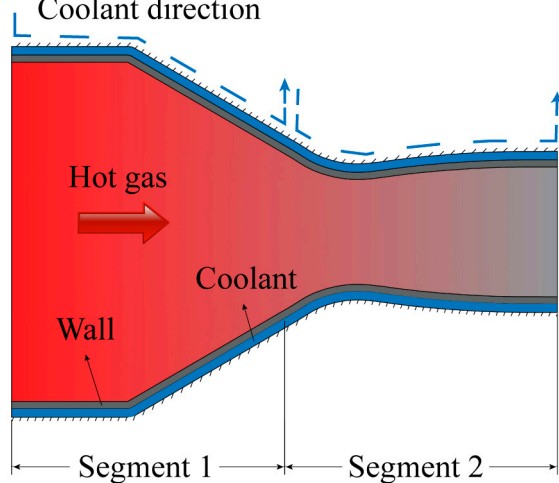

**Figure 4.** The configuration of the AEDC nozzle.

As one of the classic examples of coupled heat transfer, many researchers have calculated this case. Some researchers used empirical formulas to consider the effect of cooling water boiling on heat transfer at the wall [41], while others ignored the effect of local boiling [42]. From the results of these researchers, it can be seen that the influence of local boiling on the calculation results is not significant. In this paper, the single-phase code is used to calculate the flow in the cooling channel. Only one working condition in which the coolant in the cooling channel may be pure liquid is numerically simulated. The dissociation caused by high temperature and the radiation of gas in the nozzle is not considered.

Figure 5 shows the grid of the nozzle. The quantity of cells was 71,282. The area with a circumferential direction of 2° of the nozzle was selected as the computational domain. The calculation parameters are shown in Table 2.

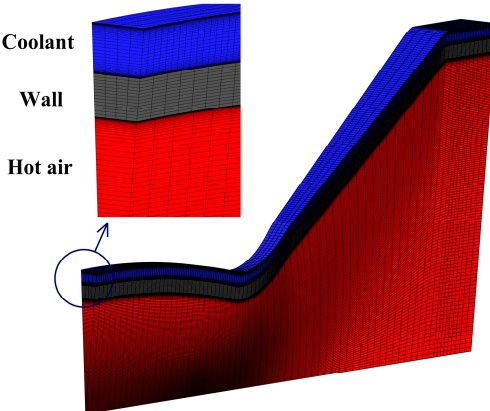

**Figure 5.** The grid of the AEDC nozzle.

**Table 2.** Calculation parameters of the AEDC nozzle.

| Air Total Pressure (atm) | Air Total Temperature (K) | Water Mass Flow Rate (kg/s) | Water Inlet Temperature (K) |
|---|---|---|---|
| 126.5 | 5000 | 5.234 | 309 |

Figure 6a shows the changes in the heat release on the hot side and the heat absorption on the cold side, monitored during the coupled heat transfer iterative calculation process. The initial overestimation of heat release on the hot side and underestimation of heat absorption on the cold side can be seen. As the calculation progresses, a negative feedback mechanism begins to take effect, causing the wall temperature to rise, resulting in an increase in heat absorption on the cold side and a decrease in heat release on the hot side. Finally, the equilibrium state is reached, and the heat released is equal to the heat absorbed.

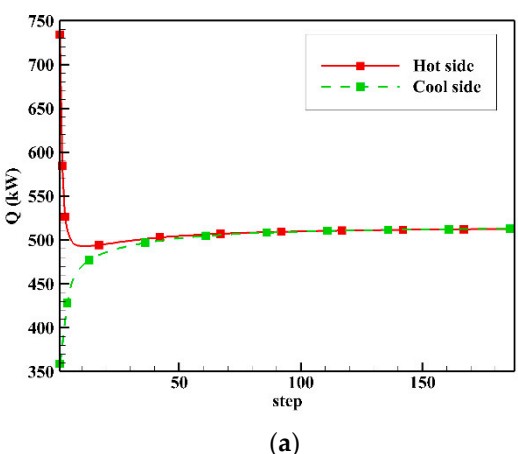

(**a**)

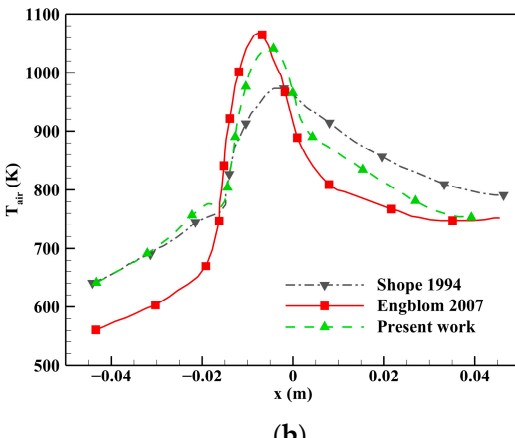

(**b**)

**Figure 6.** (**a**) Heat flow convergence curve; (**b**) Wall temperature at the throat, including a comparison of the calculation results in this paper with reference [40,41].

Figure 6b shows the temperature distribution along the hot side and cold side of the structure after the iterative calculation of the coupled heat transfer of the AEDC nozzle converges. The calculation results of this method showed good consistency with the results of other researchers. Table 3 lists the test values and calculated values of the cooling water temperature rise in the second cooling channel of the nozzle. The temperature rise in water in the calculation result is 12.7 K, which is slightly lower than the temperature rise measured in the test, 13.9 K [40]. Because the error is very small, it can be considered that the result is in line with reality.

**Table 3.** Coolant temperature rise.

| Experimental Temperature Rise (K) | Calculate Temperature Rise (K) |
|---|---|
| 13.9 | 12.7 |

### 4.2. Composite Cooling Liquid Rocket Engine

In this section, the coupled heat transfer calculation and analysis are performed for a typical liquid rocket engine. The engine has a length of 613 mm, a combustion chamber radius of 44 mm, a throat radius of 15 mm, and a nozzle outlet radius of 150 mm. The cooling mode of the liquid rocket engine is shown in Figure 7. The fuel and oxidizer, respectively, are UDMH and NTO, and the thrust chamber structure material is stainless steel. The thermal protection of the thrust chamber wall of the engine is mainly active cooling. A composite cooling method combining regenerative cooling, film cooling, and radiation cooling was adopted. Regenerative cooling was adopted in most of the combustion chamber and nozzle, in which fuel is used for regenerative cooling near the combustion chamber and throat, and the spiral channels were adopted. The oxidant was used for regenerative cooling in the middle part of the nozzle expansion section. The extension section of the nozzle adopted a relatively simple form of radiation cooling. Other parameters of the thrust chamber are shown in Table 4.

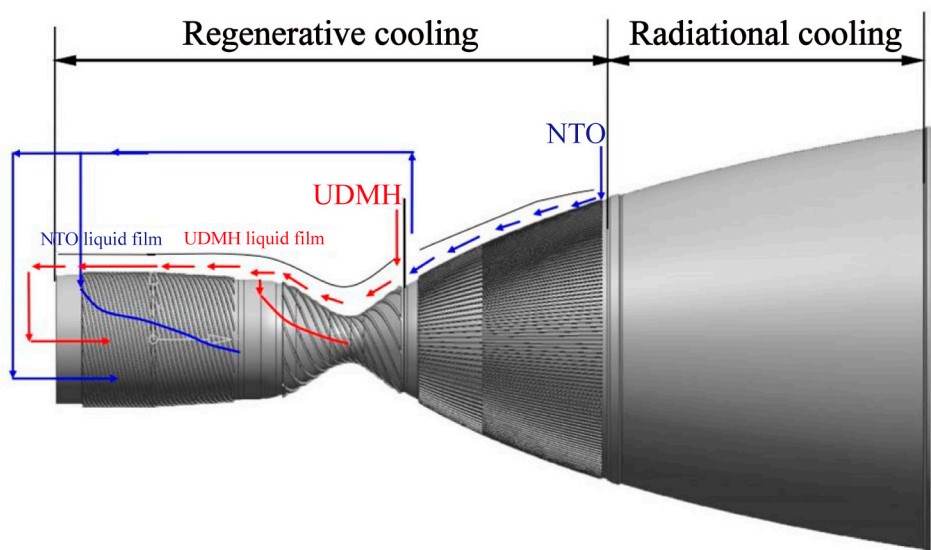

**Figure 7.** The cooling method of the liquid rocket engine.

**Table 4.** Parameters of the thrust chamber.

| Propellant | Thrust Chamber Pressure (MPa) | Mass Flow Rate (kg/s) | Cooling Channel Inlet Temperature (k) | Film Cooling Flow Rate (kg/s) |
|---|---|---|---|---|
| UDMH | 5.15 | 0.621 | 284.35 | 0.0994 |
| NTO | | 1.405 | 284.55 | 0.3232 |

This engine has a large size and a high combustion chamber temperature, resulting in a very short liquid film length. Therefore, we simplified the process of spray atomization and liquid film cooling, treating all propellants entering the combustion chamber as gas. At the location of the gas film injection point in the flow field, we added the gas film to the source term of Equation (1) based on the mass flow rate, velocity, and other parameters of the gas film. To reduce computational complexity, only one channel was used for each cooling channel section for calculation. Figure 8a,b show the grids of typical spiral and straight cooling channels, respectively. The blue area in the figure represents the fluid grid, while the gray area represents the solid grid. Only the 2° area in the circumferential

direction of the thrust chamber was taken for the calculation. The flow field grid in the thrust chamber is shown in Figure 9a. Periodic boundary conditions were used on both sides of the area in the circumferential direction. Three sets of grids were used to verify the grid independence of the flow field in the thrust chamber, with the number of grids increasing in sequence. Figure 9b shows the heat flux along the wall computed by using the three different grids. Mesh 3 was adopted in this study.

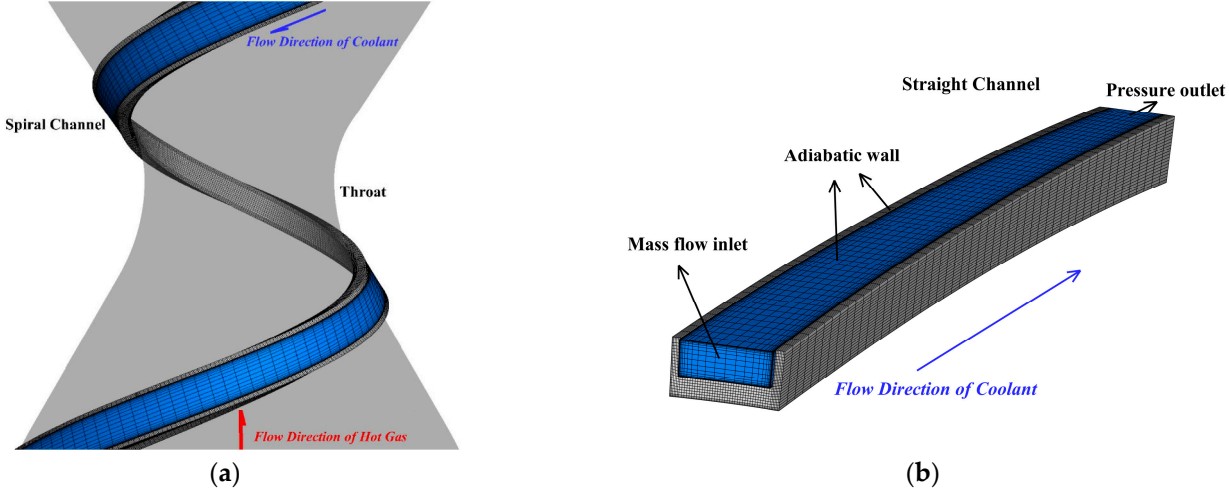

**Figure 8.** (**a**) Grid of spiral cooling channel at the throat; (**b**) Grid of straight cooling channel at the expansion section.

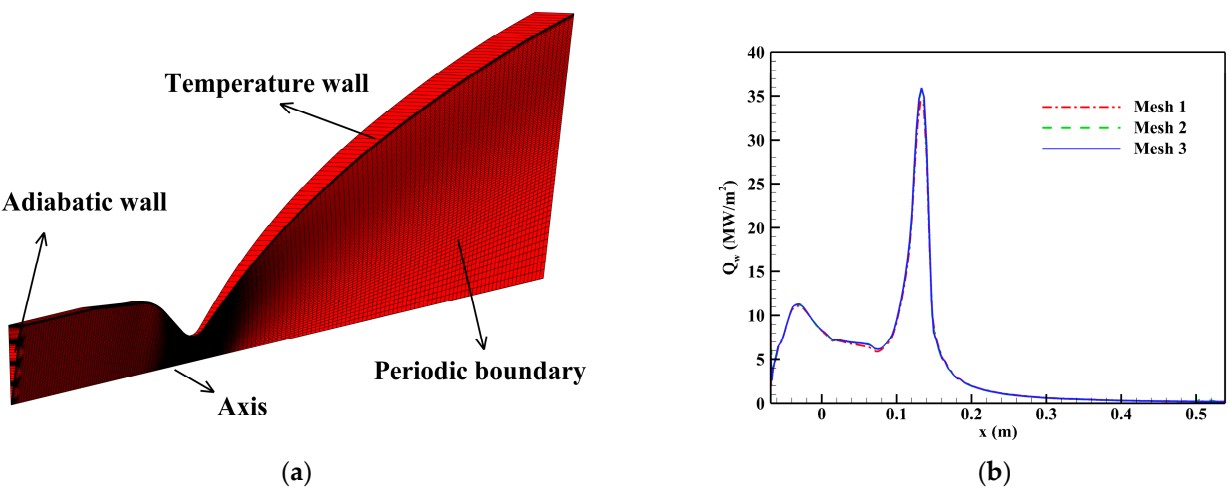

**Figure 9.** (**a**) The grid of flow field in the chamber; (**b**) Heat flux along the wall.

A coupled heat transfer calculation (maintain equal fuel mass flow rate) was carried out for two working conditions: with and without film cooling. The total heat flow of the hot side and cold side monitored in the iterative calculation of coupled heat transfer is shown in Figure 10a,b. When the heat absorption of the gas-side wall is equal to the heat release of the cooling-channel-side wall and does not change for some time, the coupled heat transfer calculation converges. When film cooling is used, the total heat flow on the wall of the thrust chamber is 0.26 MW, which is far lower than that without film cooling (0.542 MW). The latter is almost twice the former, which reflects that the cooling effect of the combined cooling method of regenerative cooling and film cooling is stronger than that of regenerative cooling only, which is in line with our expectations.

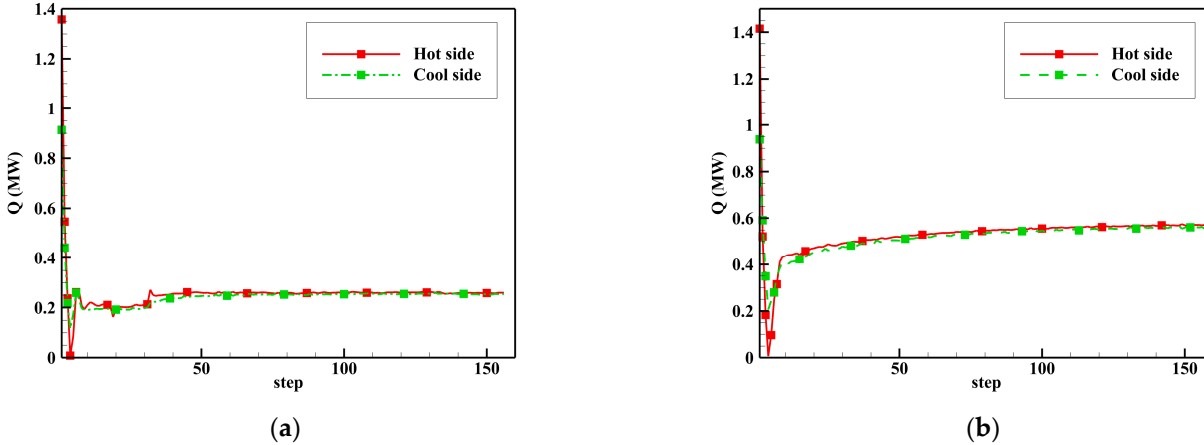

**Figure 10.** Heat flow (**a**) with gas film; (**b**) with no gas film.

Figure 11a shows the temperature distributions in the flow field with/without gas film. The flame is relatively long. The flame peak is mainly located in the second half of the combustion chamber, and the flame temperature is about 3000~3400 K. The combustion mechanism of UDMH/NTO is relatively complex. In the literature we consulted, we did not find the reaction mechanism of UDMH/NTO. Therefore, the chemical equilibrium hypothesis was adopted in the combustion simulation in this paper. Figure 11b shows the UDMH concentration near the gas film injection location of UDMH. The high concentration and low temperature of UDMH near the wall have a strong protective effect on the wall. The temperature distribution of the flow field near the gas film injection point also significantly decreases.

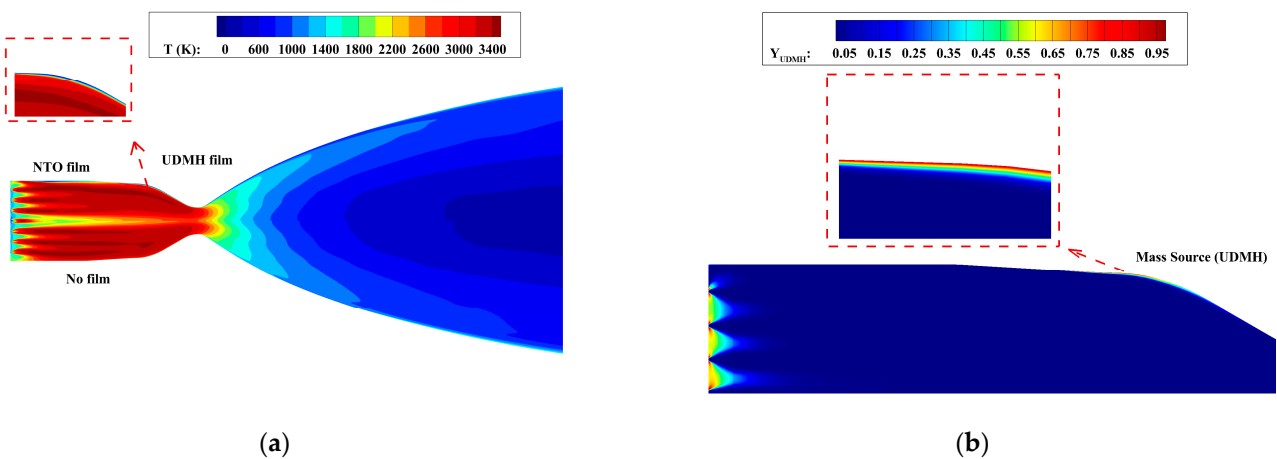

**Figure 11.** (**a**) Temperature distribution of thrust chamber flow field (with or without gas film comparison); (**b**) Mass fraction of UDMH near the injection location of film cooling.

Figure 12 shows the temperature and heat flux distribution along the gas side wall. The dashed line in the figure is the wall profile of the combustion chamber. As can be seen from the figure, when composite cooling is adopted, the peak value of wall heat flux is 14.8 MW/m$^2$, located near the throat, where the corresponding temperature is 813 K, far lower than the allowable temperature of stainless steel. In addition, there are two local minimum regions of temperature and heat flux near the head of the combustion chamber and the nozzle throat. This is because NTO and UDMH are injected into the combustion chamber from the cooling rings in these two regions, respectively, forming a local low-temperature protective layer. Only radiation cooling was adopted in the nozzle extension section; the temperature rose to 1300 K, which is close to the allowable temperature of

stainless steel. Compared with composite cooling, when only regenerative cooling is used, the wall temperature and heat flux greatly increased. The wall temperature reached more than 1000 K, and the throat heat flux could reach 27.8 MW/m$^2$. In the design condition of the engine, simple regenerative cooling cannot meet the demand for thermal protection, so the methods of film cooling and regenerative cooling must be combined. In the combustor region, when no film cooling was carried out, the wall temperature and heat flow near the jet plate were both higher, which is because the temperature of the regenerated coolant gradually increased, and the cooling effect decreased. After using the gas film cooling, the wall temperature and heat flux gradually increased after reaching the lowest point near the jet plate, which is caused by the increase in the gas film temperature. It can be seen that the film cooling changes the distribution of temperature and heat flux in the combustion chamber area.

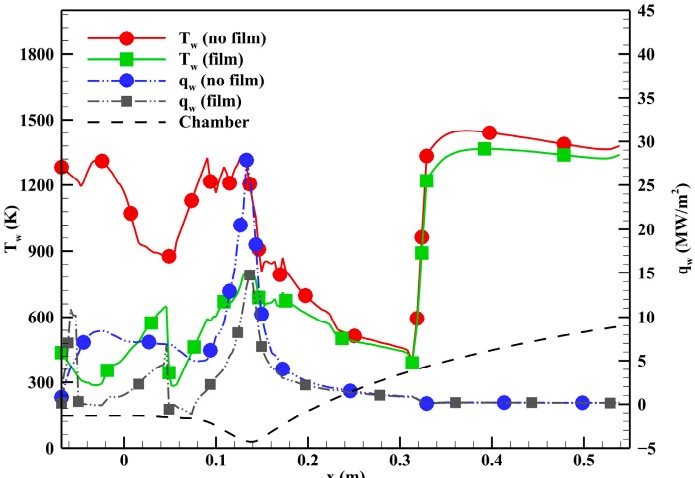

**Figure 12.** Temperature and heat flux distribution along the hot side.

Figure 13a, respectively, shows the convective heat flux and radiant heat flux in the thrust chamber. The radiant heat flux is relatively large in the high-temperature area in the chamber, but it is still far smaller than the convective heat flux, and its contribution to the total wall heat flux only accounts for 10.5%. To evaluate the efficiency of the film cooling, the non-dimension film cooling effectiveness $\eta$ is defined by:

$$\eta = \frac{T_{wg,0} - T_{wg,film}}{T_{wg,0} - T_{film,inlet}} \tag{19}$$

where $T_{wg,0}$ is the wall temperature in the absence of the gas film, $T_{wg,film}$ is the wall temperature in the presence of the gas film, $T_{wg,inlet}$ is the gas film inlet temperature. The gas film cooling efficiency of the thrust chamber is shown in Figure 13b. The film cooling efficiency is very high near the gas film injection point. In some areas far from the injection point, the film cooling efficiency decreases rapidly. Therefore, in engines with large volume and high temperature, it is necessary to use multiple film cooling circumferential seams to improve the thermal protection effect.

Figure 14 shows the temperature distribution of the cooling channel structure and the internal coolant of the nozzle expansion section. In addition, it can be seen from the figure that the structure temperature corresponding to the entrance of the cooling channel is lower. The temperature distribution of the coolant at the exit of the cooling channel shows an obvious thermal boundary layer effect. The temperature of the fluid close to the wall in the cooling channel is the same as that of the wall, so the continuity of temperature in the coupling process is maintained well.

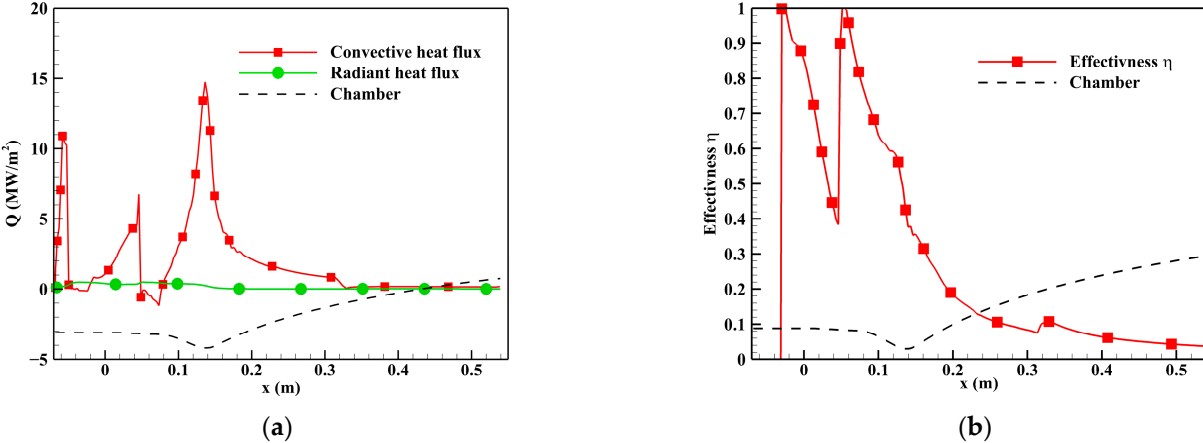

**Figure 13.** (**a**) Convective heat flux and radiant heat flux (film cooling); (**b**) Film cooling efficiency.

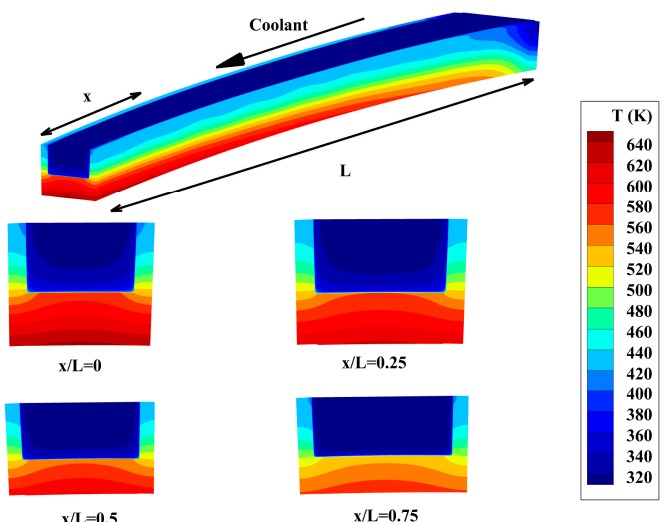

**Figure 14.** The temperature of a certain section of wall and coolant.

Figure 15 shows the temperature of the coolant under two conditions. Under the protection of gas film, the outlet temperature of both propellants is low. However, if there is no gas film, the temperature rise in UDMH used for regenerative cooling on the combustor wall can reach 223.52 K, and gasification occurs in the cooling channel. For hypergolic propellant, it is not expected to gasify in the cooling channel.

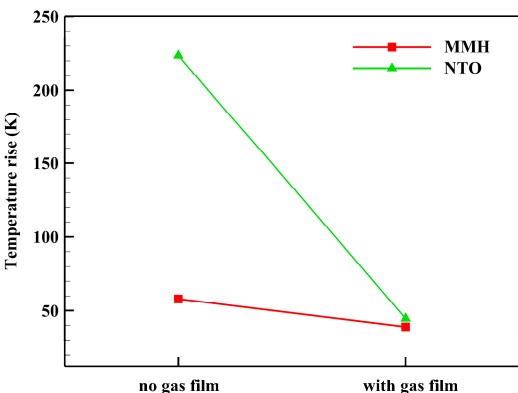

**Figure 15.** The temperature rise in coolant under different operating conditions.

## 5. Conclusions

The coupled heat transfer strategy was used to simulate the AEDC high-enthalpy nozzle and a liquid rocket engine, both of which can reach convergence state quickly. The coolant temperature rise in the numerical simulation results was in good agreement with the experimental measurement, which proves the effectiveness of the algorithm. The algorithm can be used to solve the fluid–structure coupled heat transfer of a liquid rocket engine. It is also applicable to other physical processes where the wall is heated and cooled at the same time.

This algorithm was used to calculate the coupled heat transfer of a liquid rocket engine using a UDMH/NTO propellant. The results show that the low-temperature protective layer of propellant near the wall can be formed by film cooling, which can greatly reduce the wall temperature and heat flux. The inner wall temperature of the throat area decreased from over 1000 K to 813 K, and the peak heat flux decreased from 27.8 MW/m$^2$ to 14.8 MW/m$^2$. Film cooling changes the distribution of temperature and heat flux in the combustion chamber area. The protection of the gas film reduced the temperature rise in the propellant in the cooling channel in the combustor region by 178.87 K, preventing the occurrence of large-scale gasification.

**Author Contributions:** Conceptualization, B.X. and X.X.; Methodology, B.C.; Validation, J.P.; Investigation, W.Z.; Writing—original draft preparation, B.X. and X.X.; Writing—review & editing, B.C., J.P. and W.Z.; All authors have read and agreed to the published version of the manuscript.

**Funding:** This research received no external funding.

**Data Availability Statement:** Not applicable.

**Conflicts of Interest:** The authors declare no conflict of interest.

## Appendix A

Eucken's formula:

$$\kappa_i = \frac{\mu_i R_0}{W_i}\left(c_{pi}\frac{W_i}{R_0} + \frac{5}{4}\right) \tag{A1}$$

where $\kappa_i$ is the thermal conductivity coefficient, $\mu_i$ is the dynamic viscosity coefficient, $W_i$ is the molar mass, $R_0$ is the universal gas constant, and $c_{pi}$ is the specific heat at constant pressure.

Lennard-Jones formula:

$$\mu_i = 2.6693 \times 10^{-6}\frac{\sqrt{W_i T}}{\sigma_i^2 \Omega_{\mu_i}} \tag{A2}$$

where $\mu_i$ is the dynamic viscosity coefficient, $W_i$ is the molar mass, $T$ is the static temperature, and $\sigma_i$ is the Lennard-Jones collision diameter and its unit is angstroms. $\Omega_{\mu_i}$ is the collision integral and its expression is:

$$\Omega_{\mu_i} = 1.147(T^*)^{-0.145} + (T^* + 0.5)^{-2} \tag{A3}$$

where $T^*$ is the reduced temperature and its expression is:

$$T^* = \frac{k_B T}{\varepsilon_i} \tag{A4}$$

where $k_B$ is the Boltzmann constant, and $\varepsilon_i$ is the Lennard-Jones well depth.

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
