# Peer review of "A Coupled Heat Transfer Calculation Strategy for Composite Cooling Liquid Rocket Engine"

_aerospace, doi:10.3390/aerospace10050473_

Round 1
Reviewer 1 Report
General comments:
The article describes a loose coupling scheme between fluid and solid for heat transfer calculations. The algorithm is validated with the solution of a reference problem and compared with other numerical results, and then it is used to solve and analyze the temperature distribution of a liquid rocket engine.
The governing equations for the different modules involved in the simulation are given. The coupled strategy and the data transfer between non-matching meshes are presented, but the author does not validate the strategy and data transfer with a simple case. They simulate a high-enthalpy nozzle and compare the wall temperature at the throat, which roughly matches the numerical reference results. After that, they solved a more complex liquid engine nozzle, obtaining the wall temperature and fluxes. Also, they simulated the effect of the film cooling, but no details are given about how it was introduced in the simulation process. It is not clear what the original contribution of the article is. What are the benefits of this algorithm when compared with others in the literature?
Specific comments:
Title: A fluid-structure coupled heat transfer calculation strategy for liquid rocket engine"S"
Comments about the title: The word "fluid-structure" is used in the title, but only solid heat conduction is solved, not structural deformation/interaction, so I recommend rewriting it to be more explicit about the scope of the paper.
Pag.1:
L13: the method of parameter transfer -> the data is transferred between meshes, not the parameters.
L20: What is the meaning of "relatively reasonable"?
L41: ...which is a typical fluid-structure coupled heat transfer problem. It is a fluid-solid heat transfer problem.
Pag 3:
L110: No mention is made about the algorithm or discretization for the gas flow in the chamber.
Pag 4:
L155: Euken and lennard-jone formulas should be added.
L166 to 175: Undefined acronyms are used without their definitions.
Pag 5:
L177: Structural heat conduction -> Solid heat conduction
Pag 6:
L225: Parameter transfer policy -> Data transfer scheme
L226 to 228: The paragraph should be rewritten.
L245: Parameter transfer policy between codes. -> Data transfer between codes.
L237 to 240: The paragraph is difficult to understand. Rewrite.
Caption Fig. 1 - Rewrite: Data transfer between codes.
L250: Number of grids? Do you mean element size?
Pag 8:
L291: AEDC should be defined when it is first used.
L294: No information about the numerical setup, grid sizes, or initial conditions are given. Is the problem solved in 2D, axisymmetric or 3D?
Pag 9:
Fig.4 What is the meaning of a segmented and continuous nozzle?
Pag 10:
L324: No reference to the experimental data is given.
Pag11:
L352: Any detail about the meshes(solid, fluid, etc) used in the simulations is given.
Pag14:
L439: Only the experimental coolant temperature rise is given.
English should be checked carefully.
Reviewer 2 Report
the described methodology is typical and there are no elements of scientific novelty in it. there are also no errors within the framework of well-known and widely used methods. In this regard, the author needs to specify specifically what is the element of scientific novelty in this article. In particular, when modeling flows in the boundary layer, the presented RANCE equations are not suitable for high-temperature differences between the wall and the potka core. This is well known from the publications of the 1980-1990 years of the last century.
Minor editing of the English language is required
Round 2
Reviewer 1 Report
The article has been edited by the authors according to the suggestions made in the previous review.
The english style can be improved
Reviewer 2 Report
Accept in present form
Accept in present form